# HD-SiO_2_/SiO_2_ Sol@PDMS Superhydrophobic Coating with Good Durability and Anti-Corrosion for Protection of Al Sheets

**DOI:** 10.3390/ma16093532

**Published:** 2023-05-05

**Authors:** Ruohan Xia, Bing Zhang, Kousuo Dong, Yao Yan, Zisheng Guan

**Affiliations:** College of Materials Science & Engineering, Nanjing Tech University, 30 South PuZhu Road, Nanjing 211816, China

**Keywords:** superhydrophobic, SiO_2_, surface modification, anti-corrosion

## Abstract

Superhydrophobic coatings with excellent water-repellent properties imply a wide range of application areas. However, improvements are needed in terms of stability and complex processing procedures. In the present study, a superhydrophobic coating on Al sheets was prepared by mixing hexadecyltrimethoxysilane (HDTMS)-modified SiO_2_ nanoparticles and acid-catalyzed silica sols (HD-SiO_2_/SiO_2_ Sol) with polydimethylsiloxane (PDMS) binder. The HD-SiO_2_ nanoparticles and acid-catalyzed silica sol (SiO_2_ sol) form a binary graded micro-nanostructure, providing excellent superhydrophobicity (Water Contact Angle = 158.5°, Sliding angle = 0°). Superhydrophobic coatings with excellent water-repellent properties have potential for corrosion prevention. However the commonly used organic resins have poor chemical and mechanical properties. In the present study, the results of outdoor exposure for 30 days, immersion in acid and alkaline solutions for 24 h, grit abrasion, and water impact experiments, respectively, showed that the prepared superhydrophobic coating has good wear resistance. The integrated superhydrophobic coating on the Al sheets exhibited good corrosion inhibition with an efficiency (η) of 98.9%, which is much higher than that of the uncoated sheets. The present study provides a promising approach for producing stable superhydrophobic coatings at a low cost, with the potential to supplant conventional organic resin anti-corrosion coatings.

## 1. Introduction

Corrosion of metallic materials is a serious threat to structural safety, life safety, environmental protection, and economic development [1,2,3,4,5]. The most common method of corrosion protection is a coating that acts as a barrier to isolate the metal from the corrosive medium [6,7,8]. A corrosive medium is a substance, such as a liquid or gas, that can chemically react with a material and cause it to deteriorate over time. Among the anti-corrosion methods in the oil and gas industry [9], organic coatings have been extensively adopted as an effective method for preventing corrosion and as a physical barrier between metallic materials and corrosive media [10,11,12]. Conventional organic coatings can be readily permeated by water molecules and corrosive agents such as acids, bases, and salts [13], resulting in the formation of conductive channels, which can exacerbate metal corrosion by facilitating electrolyte-metal surface interaction [14,15,16]. However, superhydrophobic coatings can effectively resist corrosion and extend the service life of metallic materials [17,18,19,20,21]. Such coatings are considered to be an effective means of corrosion protection and have become increasingly popular [22,23].

Superhydrophobic materials have a water contact angle (WCA) greater than 150° and a sliding angle (SA) less than 10° [24]. Superhydrophobic surfaces can be widely found in nature, such as lotus leaves [25], fly eyes [26], butterfly wings [27], and cicada wing surfaces [3]. Such waterproof materials are biotically prepared for applications in self-cleaning [28], anti-icing [29], oil and water separation [30,31], drag reduction [32], and corrosion protection [33,34,35]. Superhydrophobic surfaces prepared by means of the traditional single-molecule layer [36] modification techniques have failed in local areas due to the “weak bonding” between other surface energy molecules and rough substrates, which greatly limits their service life and scope [37,38,39].

The purpose of this paper is to investigate a method for preparing superhydrophobic coatings and to evaluate their performance. In the study, the superhydrophobic material was characterised using FT-IR, SEM, and AFM techniques in order to analyse its properties. Subsequently, the coating was applied to Al sheets by spraying methods and its performance was analysed experimentally when subjected to water impact, grit abrasion, and acid/alkali immersion in order to assess its stability. In addition, corrosion resistance tests were carried out to examine the corrosion resistance of the coating. The results of the study show that the prepared superhydrophobic coatings have excellent superhydrophobicity, stability, abrasion resistance, and corrosion resistance. This study is therefore a guide to the development of high performance superhydrophobic coatings and may provide a reference for applications in related fields.

## 2. Experimental

### 2.1. Materials

Al sheets (65 mm × 25 mm × 2 mm) were purchased from local markets; SiO_2_ (30 nm) was purchased from Aladdin Reagent Co. (Shanghai, China); Sylgard 184 organ polydimethylsiloxane (PDMS) and its curing agent were supplied by Dow Corning (Midland, TX, USA); hexadecyltrimethoxysilane (HDTMS, ≥85%) was obtained from Aladdin Reagent Co. (Shanghai, China); isopropyl alcohol, tetraethoxysilane (TEOS) was obtained from Sinopharm Reagents Co. (Beijing, China); hydrochloric acid (HCl) was supplied by Shanghai Lingfeng Chemical Reagent Co. (Shanghai, China); and deionized water was home-made from a laboratory ultrapure water machine (UP-3L). All chemical reagents did not require further purification.

### 2.2. Pre-Treatment of Substrate Materials

To remove oil and impurities from the surface, the substrate materials were ultrasonicated in deionized water for 10 min and then in ethanol for another 10 min. Subsequently, the materials were rinsed with deionized water and then ready for use after being dried in an oven.

### 2.3. Preparation of HD-SiO_2_/SiO_2_ Sol@PDMS Superhydrophobic Coatings

A typical preparation process is shown in Figure 1. Firstly, 0.4 g of 30 nm SiO_2_ was dispersed in a water solution (the volume ratio of isopropanol to water was 10:1), then an appropriate amount of hydrochloric acid was added to adjust the pH to 3. Secondly, 1.5 mL of TEOS and 0.5 mL of HDTMS were added slowly to the solution in sequence. After stirring at room temperature for 3 h, a homogeneous milky white solution of modified SiO_2_ and silica sol was obtained (the pH value of the solution increased and remained unchanged for a long time). Thirdly, 2 wt.% PDMS was added dropwise to the sols and stirred for 30 min to obtain a HD-SiO_2_/SiO_2_ Sol@PDMS mixture. Next, the mixture was sprayed onto the surface of an Al sheet using a 0.2 Mpa gun to achieve cross-linking and curing of the coating. Finally, the HD-SiO_2_/SiO_2_ Sol@PDMS coating was obtained by continuous drying at 100 °C for one hour. The thickness of all coatings was controlled at 25 ± 5 µm via spraying.

### 2.4. Sample Characterization

The surface morphology was observed by means of field emission scanning electron microscopy (FE-SEM, Zeiss Gemini 300). In order to improve the electrical conductivity, the specimens were coated with a considerably thin layer of gold prior to testing. A microscopic area of the sample surface scale (5.0 μm × 5.0 μm, 2.5 μm × 2.5 μm) was observed using an atomic force microscope (AFM, Bruker (Bruker, Billerica, MA, USA)) to obtain structural knowledge for analysis of the effect of roughness on the wettability of the coating. To establish the chemical composition and analyze the principles of coating preparation, the chemical functional group on the surface of the sample was analyzed by means of Fourier transform infrared spectroscopy (FT-IR, Nicolet Nexus 670, 4000~400 cm^−1^, 4 cm^−1^, 32, KBr pellet method).

### 2.5. Electrochemical Testing

The corrosion resistance of the coated Al sheets was investigated using a CHI604E electrochemical workstation with a three-electrode system (a graphite electrode for the auxiliary electrode, a saturated glycolic electrode for the reference electrode, and a working electrode). The samples were placed in 3.5 wt.% NaCl solution and the test was started after the open circuit voltage (OCP) stabilized. The dynamic polarization curves of the specimens were registered at a rate of 1 mV/s. The electrochemical impedance spectra (EIS) with an amplitude of 10 mV were tested within the range from 0.1 to 10^5^ Hz. All the electrochemical parameters were simulated by means of ZSimpwin software (3.20, Hermann W. Rometsch: Ludwigsburg, Germany).

## 3. Results and Discussion

### 3.1. Surface Morphology & Self-Cleaning

In order to explore the influence of each component of the coating on the hydrophobicity of the coating, the control variable method was used to study the influence of each component on the coating.

The content of SiO_2_ obviously affected the hydrophobicity of the coating, as shown in Figure 1a. According to the experimental plan in Section 2.3, the ingredient table is shown in Appendix A. The WCA increased with the increasing SiO_2_ content, and the WCA of 158.5° was obtained at 0.8 wt.% and 1.0 wt.% SiO_2_. This indicates a close relationship between the wettability of the coating and the content of SiO_2_. To further investigate the coating, the microstructure of different contents was observed, as shown in Figure 1b–g. With the increase in SiO_2_ content, the micro-rough structure on the surface of the coating increases significantly, which allows the surface to capture more air and form an air film, hindering direct contact between water and the coating surface and improving the hydrophobicity and anti-corrosion properties of the coating surface. Thus, incorporating 0.8 wt.% SiO_2_ nanoparticles is highly advantageous for improving superhydrophobicity.

The content of tetraethoxysilane (TEOS) as a precursor of silica sols had significant effects on the shape and stability of the sol particles. Figure 2a shows the WCAs and SAs of the coatings prepared at different TEOS contents. According to the experimental plan in Section 2.3, the ingredient table is shown in Appendix A. Evidently, the WCAs decreased and the SAs increased significantly when the TEOS content was higher than 2 wt.%. Acid-catalyzed silica sols were prepared using the classical Stober method [40,41], whereby TEOS hydrolysis produced silanol that condensed in any direction under acidic conditions, eventually forming a linear chain-like silica structure consisting of a large number of small particles. Once the TEOS content was excessive, acid-catalyzed silica sols tended to form spherical particles with larger sized particles due to faster condensation hydrolytic reactions. The results obtained were a loose coating and decreased hydrophobicity. In order to achieve ideal coating structures, TEOS content needs to be strictly controlled. The best hydrophobicity achieved was at a TEOS content of 2 wt.%. As a binder, PDMS also play an important role in determining hydrophobicity and stability of the coatings. Figure 2b shows SEM images and illustrations of 0.8 wt.% SiO_2_ nanoparticles dispersed in different PDMS. According to the experimental plan in Section 2.3, the ingredient table is shown in Appendix A. SEM results indicate that when the PDMS content is 5 wt.% (Figure 2(bI)), there are much fewer nanopores than at 2 wt.% (Figure 2(bII)) and 1 wt.% (Figure 2(bIII)). The result is that the WCA (135°) is also lower than in other samples (158.5°, Figure 2(bII)) and has a content of 1 wt.% (158.5°, Figure 2(bIII)). Conversely, when the HD-SiO_2_/SiO_2_ Sol@PDMS coating was prepared using 1 wt.% PDMS, the majority of the nanoparticles were located on the coating surface, providing sufficient roughness to ensure superhydrophobic properties. Further research will be conducted on the formulation of 0.8 wt.% SiO_2_, 2 wt.% TEOS, and 2 wt.% PDMS, as it has demonstrated the highest hydrophobicity (WCA = 158.5°) and stability among all tested formulations. The formulations for further research will be referenced from Appendix A.

The typical morphology of the HD-SiO_2_/SiO_2_ Sol@PDMS superhydrophobic coating was observed using SEM and AFM, as shown in Figure 3a,b. Photographs of different droplets (HCl, NaOH, juice, coffee, and H_2_O) on the coating reveal that the droplets could be maintained as near-perfect spheres. The SEM image shows that the surface of the coating has a papillary structure similar to a lotus leaf, and many prominent nanoparticles could be observed on the surface of the papillary structure (Figure 3a). The notable superhydrophobic properties of the coating could be attributed to a binary micro-nano structure consisting of a composite product of SiO_2_ and acid-catalyzed silica sols. The roughness of the coating is further analyzed using AFM, which reveals that it is non-flat, with micro-nano bumps and depressions. The microstructural height range of the lowest peak was 280.4 nm and the highest peak was 244.4 nm, as shown in Figure 3b. Additionally, the cross-sectional profile (Figure 3b) shows fluctuations between peaks and valleys ranging from 0.5 μm to 1.5 μm. The results indicate that the coating was in the superhydrophobic state, where an air layer is formed when water and the coating come into contact. Such states can reduce the contact between the liquid and the material surface. The self-cleaning properties of the coating were investigated by simulating the source of contamination with soil and sand. Figure 3c shows the photographs of a water droplet before and after rolling on the superhydrophobic coating. The water droplet appears to roll downwards in a spherical shape and remove surface impurities, indicating the coating has good self-cleaning properties.

### 3.2. Chemical Composition

FT-IR was applied to analyze the chemical composition of HD-SiO_2_/SiO_2_ Sol@PDMS superhydrophobic coating. As shown in Figure 4, there are two characteristic vibrational peaks at 803 and 902 cm^−1^, which can be attributed to the anti-symmetric stretching vibration of Si-O-Si and the networked Si-O-Si symmetric bond stretching vibration in the coating [42]. The successful binding of the nano-silica to the acid-catalyzed silica precursors formed a networked silica structure. The characteristic peak at 1119 cm^−1^ in the figure is the absorption peak of the Si-C bond. Further, two characteristic peaks at 2961 and 2921 cm^−1^ were also observed, which could be attributed to bending and telescopic vibrations of aliphatic-CH_2_ associated with the PDMS contained in the coating as well as the hydrophobic modifier HDTMS [43]. The presence of these peaks, when compared to the infrared spectrogram in Appendix A, further confirms the successful preparation of the HD-SiO_2_/SiO_2_ Sol@PDMS superhydrophobic coating.

### 3.3. Stability

The mechanical durability of superhydrophobic coating is a key focus in practical applications. Hierarchical micro/nanostructures have a tendency to lose their superhydrophobic performance when subjected to mechanical friction. Hence, grit abrasion testing was conducted on the HD-SiO_2_/SiO_2_ Sol@PDMS coating to evaluate the mechanical durability. Figure 5a illustrates the procedure of immersing the coating in a beaker containing standard sand followed by its removal to constitute a single wear cycle. Note that a force is applied to the coating here to complete a cycle at a speed of 20 cm/s. The WCA and SA of the coatings were measured after five wear cycles. Figure 5b displays the WCAs and SAs of the coating after grit abrasion testing. The results demonstrate WCAs decreased and SAs increased with increasing wear cycles; however, the WCA still remained above 150° after grit abrasion over 50 cycles. These results demonstrate that the HD-SiO_2_/SiO_2_ Sol@PDMS coating has good wear resistance. Subsequently, the strength and adhesion of the coating to the substrate were evaluated through water impact testing, measured at 200s intervals, as shown in Figure 5c. Obviously, the WCA of the HD-SiO_2_/SiO_2_ Sol@PDMS coating decreased from 158.5° to 151° under a water pressure shock of 200 KPa for 1200 s and still remained superhydrophobic.

For exploring the chemical stability of the HD-SiO_2_/SiO_2_ Sol@PDMS superhydrophobic composite coating, the WCAs and SAs of the coating were measured after immersion in 0.1 mol/L HCl, H_2_O, 0.1 mol/L NaOH, and 3.5 wt.% NaCl solutions for 24 h, respectively. As shown in Figure 6a, the WCA of the coating after 24 h immersion was greater than 150° and the SA was less than 10°. Notably, the superhydrophobic property of the coating obviously decreased in the alkaline environment, which could be attributed to the weak alkali resistance of the PDMS. As a result, the coating could easily react with alkaline substances leading to the destruction of its hydrophobic structure and diminishing the hydrophobic property. In order to characterize the weatherability of the coating, the coating was placed in outdoor exposure for 30 days before being tested. Here, after placing the coating outdoors, the temperature and humidity were measured three times a day (morning, noon, and evening), obtaining an average temperature of 17.3 °C and an average humidity of 77%. The results in Figure 6b show that the coating maintained superhydrophobic properties.

### 3.4. Corrosion Resistance

To investigate the ability of the HD-SiO_2_/SiO_2_ Sol@PDMS superhydrophobic coating to be applied to metal corrosion protection, electrochemical corrosion tests were performed on Al sheets that had been subjected to different surface treatments in 3.5 wt.% NaCl solution. As shown in Figure 7, the corrosion potentials were lowest for the superhydrophobic coated sample, followed by the PDMS coated sample, followed by the bare Al sheet sample. The electrochemical parameters including corrosion potential (E) and corrosion current density (I) for each sample obtained from the polarization curve fitting are shown in Table 1. The corrosion current density of the superhydrophobic coated sample was 6.11 × 10^−7^ A/cm^−2^, which was two orders of magnitude less than that of the bare Al sheet of 5.56 × 10^−5^ A/cm^−2^, demonstrating that the superhydrophobic coating significantly improved the corrosion resistance of the Al sheet. The corrosion efficiency (η) of the superhydrophobic coating sample relative to the original material could be calculated using Equation (1) [44].
(1)$$\eta \left(\%\right)=\left(1-\frac{I_{SHC\;Al}}{I_{Al}}\right)\times 100\%$$

*I_SHC Al_* and *I_Al_* represent the corrosion current densities of the superhydrophobic coated sample and the bare Al sheet sample, respectively. The results show that the corrosion inhibition efficiency of superhydrophobic coated sample reached as high as 98.9%, whereas that of the HD-SiO_2_/SiO_2_ Sol@PDMS coated sample was 86%. Due to the presence of air in the micro-nanostructure of the superhydrophobic surface, the trapped air can act as an effective barrier to prevent the corrosive medium in NaCl solution from reaching the surface of Al sheet, thereby providing good corrosion protection [45].

The chemical corrosion resistance of the samples was further evaluated by EIS, so as to reveal the detail of the metal corrosion process. The Nyquist and Bode plots of the three samples are given in Figure 8. The capacitive loop and impedance modulus of the superhydrophobic coated sample were much larger than those of the PDMS coated sample and the bare Al sheet sample, as can be obviously seen from the plots. The data obtained by fitting the circuit were consistent with the image display, and the coating resistance could be ranked in the following order: superhydrophobic coated sample > PDMS coated sample > Al sheet sample. In general, larger Nyquist rings have a higher charge transfer resistance, which indicates a lower corrosion efficiency [46]. Upon immersion of the Al sheet coated with prepared superhydrophobic surfaces into a corrosive solution, hierarchical micro/nanostructures on the superhydrophobic surface formed composite interfaces with the solution, wherein air was confined in the valleys among the hierarchical structures. The composite interfaces led to reduced contact area between the corrosive solution and the metal layer, thereby acting as a barrier against corrosion and hindering the transfer of electrons and ions between the substrate and the electrolyte (as depicted in Figure 8c). Therefore, the application of a superhydrophobic coating can provide an effective barrier coating for Al sheets.

## 4. Conclusions

The HD-SiO_2_/SiO_2_ Sol@PDMS superhydrophobic coating was prepared by compounding hydrophobically modified silica and acid-catalyzed chain silica sol to form a binary micro-nanostructure and introducing PDMS as a binder co-blended together. After exploring the effects of each component on the coating, it was finally determined that the formulation with 0.8 wt.% SiO_2_ content, 2 wt.% TEOS content, and 2 wt.% PDMS content was the superhydrophobic coating. The coating has good mechanical stability after 50 sand abrasion cycles and water impact tests. The superhydrophobic coated sample had a larger coating resistance and lower corrosion current density, it prevented the occurrence of corrosion reactions on the Al sheet surface, and it slowed down corrosion, with the superhydrophobic coating having a corrosion inhibition efficiency of 98.9%, as tested by EIS. Based on the excellent corrosion protection performance of HD-SiO_2_/SiO_2_ Sol@PDMS superhydrophobic coating on Al sheets, we believe that superhydrophobic coatings have potential applications in metal corrosion protection.

## Data Availability

Not applicable.

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
