# Peer review of "HD-SiO_2_/SiO_2_ Sol@PDMS Superhydrophobic Coating with Good Durability and Anti-Corrosion for Protection of Al Sheets"

_materials, 2023, doi:10.3390/ma16093532_

Round 1
Reviewer 1 Report
The authors report on the preparation of the HD -SiO2/SiO2 Sol@PDMS for superhydrophobic coatings on aluminum sheet. The coating exhibits good mechanical stability and improves corrosion resistance on Al sheet surface. However, there are some points that are not clear yet, as the following comments.
1. The mechanical durability of the superhydrophobic coating was tested using the grit abrasion test. How do the authors control the applied force and speed of the sample during the procedure of dipping the coating into a beaker of standard sand and then removing it to form a single wear cycle? Please explain in the text.
2. What is the average temperature and average humidity during the 30-day outdoor exposure to test the chemical stability of the superhydrophobic composite coating HD -SiO2/SiO2 Sol@PDMS? Please report in the text.
3. The authors did not mention how to calculate the corrosion rate and H(%) in Table 1 (please elaborate in the text).
Reviewer 2 Report
Reviewer comments:
General Comments
1. Please see my comments and suggestions in the marked-up copy provided.
2. The hydrophobicity and the anti-corrosive properties of the selected formulation are well characterized in this investigation, but the work needs a major revision especially with a complete description of the methods and analyses in the experimental. The methods for the various formulations and percentages for silica content need to be put in a table with data provided in the supporting information. Please see the marked up copy for more details.
3. The manuscript also needs to clearly define what ratio (%, volume or mole) has been used to characterize the various techniques. The selective use of a single coating formulation from the range tested and its characterization limits the understanding of this work.
4. If the hypothesis is to understand this methodology/strategy for the preparation of superhydrophobic coatings as claimed on line 8 of the conclusion, then all coatings should be compared to determine the most superhydrophobic. The select one cannot be called typical for all the formulations since they have not been characterized and presented even in the supporting information. Also, the SiO2 content changes and the PDMS content changes which changes the CH2 to SiO2 ratio along with the HDTMS organic content. These will not show the same morphology as the one presented in this work to indicate hydrophobicity and corrosive resistance.
5. The reason why a select coating was claimed as superhydrophobic and chosen to be presented for all the techniques to make a general conclusion for these coatings needs to be clearly stated if that is what the authors have done.
6. The FTIR needs comparative spectra with PDMS and TEOS.
7. The language in the Introduction needs to be revisited.
8. The labeling in the figures is inconsistent and needs a better representation.

Round 2
